# Contribution to Controlled Method of Varnish Removal from Easel Paintings by ns Pulsed Nd:YAG Laser

**Maxime Lopez** [1,2], **Xueshi Bai** [2], **Nicolas Wilkie-Chancellier** [1] and **Vincent Detalle** [1,2,3,*]

1 SATIE, Systèmes et Applications des Technologies de l'Information et de l'Energie, CY Cergy-Paris Université, CNRS UMR 8029, ENS Paris-Saclay, 5 Mail Gay Lussac, 95031 Neuville sur Oise, France
2 C2RMF, Centre de Recherche et de Restauration des Musées de France, 14 Quai François-Mitterrand, 75001 Paris, France
3 ITMO University, 197101 St. Petersburg, Russia
* Correspondence: vincent.detalle@cyu.fr

**Abstract:** Varnish removal from easel paintings is a restoration procedure that is regularly undertaken by cultural heritage conservators. In very few cases, traditional methods (chemical and mechanical) do not allow them to reach the goal of a perfectly controlled and selective cleaning. UV laser ablation has been considered since the 1990s to overcome these limitations, but its application to real cases is far from common practice. This study proposes a calibrated procedure based on ablation by an ns UV Nd:YAG laser at 266 nm combined with optical coherence tomography for micrometric laser varnish removal from inhomogeneous and highly photosensitive pictorial layers. The use of this wavelength for varnish removal, not adapted according to the literature, is discussed again after the beam shaping leading to a homogeneous and controlled intensity distribution. The innocuity is controlled by laser-induced luminescence, and some solutions for the transparency loss of the residual varnish following ablation are proposed.

**Keywords:** laser cleaning; cleaning of easel paintings; varnish removal; laser ablation; Nd:YAG; 266 nm; beam shaping; optical coherence tomography (OCT); laser-induced luminescence (LIL)

## 1. Introduction

The cleaning of paintings is a common conservation procedure. It encompasses the removal of dust and pollutants from the surfaces of paintings, the thinning or removal of tarnished varnishes (often followed by the application of a new varnish layer), and the removal of irrelevant overpaints or overflowing mastics. Except for dust removal, which is generally undertaken by soft mechanical action, cleaning procedures involve the use of a swab soaked with the proper liquid, which allows for the selective removal of the problematic material without jeopardizing the underlying layers. Exceptionally, a hybrid method combining the softening of the material by a liquid and mechanical action with a scalpel can be applied. Despite its invasive and irreversible nature, cleaning is regularly undertaken by painting conservators. Cleaning operations aim at restoring the readability of the artwork by bringing it closer to what is considered to be the artist's original intention. In some specific and scarce cases, these methods do not allow conservators to reach the goal of a perfectly controlled cleaning. This is the case, for example, for very fragile artworks that can barely be touched, for very oxidized materials that demand aggressive solubilizing agents (thus compromising the underlying layers), or for the removal of very thin layers, which demands tremendous precision.

To overcome these limitations, pulsed-laser cleaning (through photothermal ablation processes) has been considered and studied since the 1990s as an alternative tool not only for varnish removal [1], but also for the elimination of overpaints and mastics [2]. Despite the unique properties of pulsed lasers in terms of intensity, spatial and temporal shaping, monochromaticity, selectivity, and coherence and directionality, theoretically allowing

unmatched resolution for the selective and controlled removal of material, their use is far from a common conservation practice. The development and generalization of this technique have been hindered by problematics, such as irreversible pigment discoloration in cases of interactions between the laser radiation and exposed pictorial layer [3], together with possible photochemical modifications induced on the remaining material [4].

Research on the specific procedure of laser varnish removal began with ns excimer lasers operating in the ultraviolet (UV) range (mainly ArF 193 nm and KrF 248 nm) [1–6]. The choice of this wavelength range was motivated by the high absorption of most natural varnishes within it, guaranteeing the possibility of a laser ablation process [1]. It gave satisfactory results, and for a time, KrF lasers were considered the most appropriate source for this conservation operation [7,8]. However, probably because of their bulky nature and intrinsic safety issues (mainly the use of high voltages and poisonous halogen gases), they were gradually supplanted by Nd:YAG lasers (fundamental wavelength: 1064 nm) [9–11]. Indeed, these solid-state sources, which are extremely robust and compact, require little maintenance, and allow for the attainment of the UV range by harmonic generation, seemed more adapted to the field of instrumentation for conservation. The fourth ($4\omega$: 266 nm) and fifth ($5\omega$: 213 nm) harmonics of the Nd:YAG, which meet the absorption criterion necessary for the ablation of most natural varnishes, were therefore evaluated and compared for laser varnish removal [12,13]. Due to the intrinsic optical properties of these varnishes, $5\omega$ Nd:YAG radiation is more absorbable than $4\omega$ radiation. This implies a lower ablation threshold and lower optical penetration depth at this wavelength, which result in a better cleaning resolution and less compromising conditions for the remaining materials. In addition, it was shown that photochemical changes in the varnish only seem to appear under irradiation at $4\omega$. Finally, varnish ablation at $4\omega$ leads to the formation of microbubbles, the diameters of which increase with the fluence, generating light scattering, opacifying the residual coating, and thus compromising the artwork's readability. This effect does not seem to appear under $5\omega$ irradiation. Currently, the literature shows that the fifth harmonic of the Nd:YAG is considered the most suited source for this application [13].

In this work, we propose a controlled, calibrated procedure for laser varnish removal from paintings that involves UV laser ablation monitored by optical coherence tomography (OCT). The $4\omega$ ns Nd:YAG radiation was reevaluated based on laser beam shaping, a laser-induced-luminescence (LIL) assessment, and the proposition of a solution to the bubble formation issue induced when using 266 nm for varnish removal.

## 2. Materials and Methods

### 2.1. Optical Setups

The laser ablation of organic materials, such as varnishes, consists in focusing a short-duration (generally from μs to fs) laser pulse onto an absorbing material to induce the ejection of molecular fragments following bond breakage due to the absorption of photons [14–16]. The minimal energy needed to induce ablation depends on the pulse duration and optical properties of the material at the working wavelength. This ablation threshold is generally measured in $mJ/cm^2$. The fraction of the laser intensity profile above this ablation threshold in the interaction plane induces an ablated footprint of the laser beam.

The laser source used for this study was a Q-switched $4\omega$ ns Nd:YAG (10 ns, 50 mJ, 20 Hz, 266 nm) from Lumibird (ex Quantel-Keopsys). Figure 1 shows the intensity spatial profile of the laser as measured by the manufacturer in the near field (Figure 1a) and far field (Figure 1b).

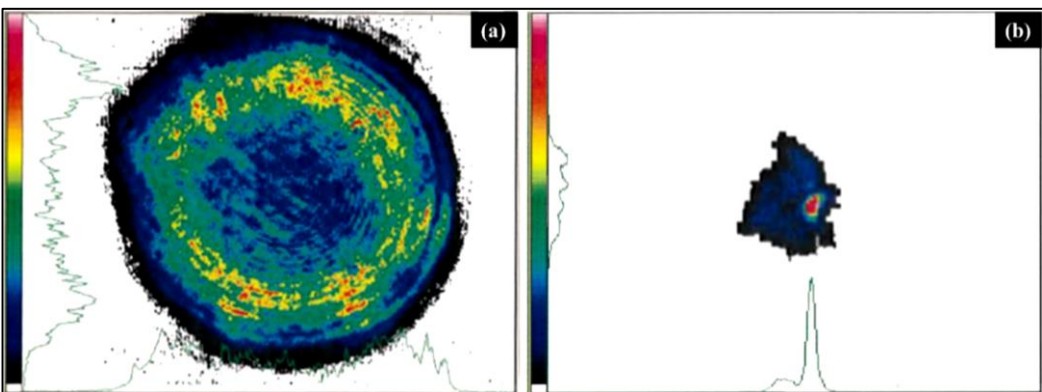

**Figure 1.** Intensity spatial profile of laser used, measured by manufacturer: (**a**) near field (output of laser); (**b**) far field (in focal plane).

Figure 2 shows the first setup (referred to as "setup 1" for the rest of the study) implemented for the laser ablation of varnish with the 4ω laser. A variable attenuator is used to finely control the laser energy, a shutter allows the selection of specific trains of pulses, and an f = 300 mm lens is used to focus the laser radiation onto the sample, which is mounted on a motorized stage. The laser beam remains fixed during the operation, and the perpendicular movement of the translation stage allows for the laser processing of a specific 2D area of the sample. A delay generator is used to synchronize the laser repetition rate, shutter, and motorized translation stage.

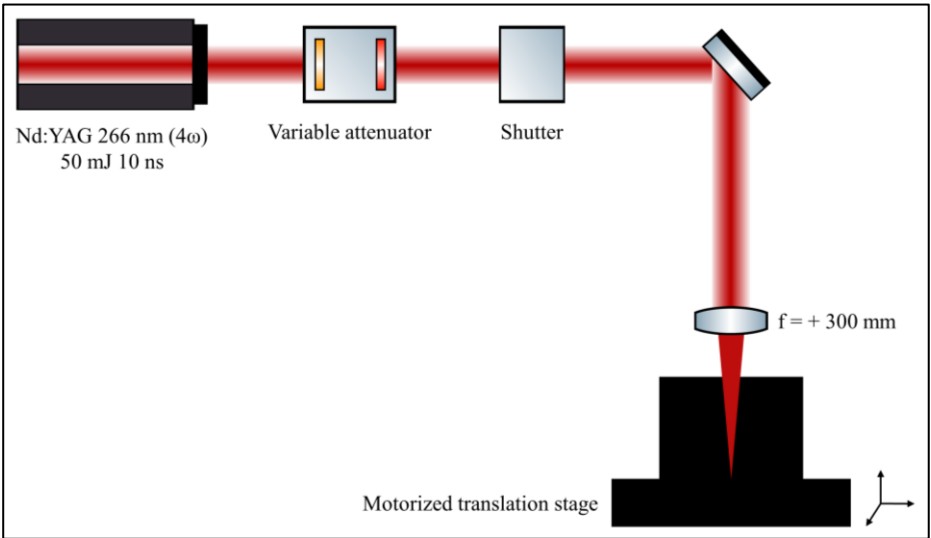

**Figure 2.** First setup implemented for laser ablation of varnish. A variable attenuator is used to control the laser energy, a shutter allows the selection of specific trains of pulses, and an f = 300 mm lens is used to focus the laser radiation onto the sample, which is mounted on a motorized stage.

To optimize the laser–matter interaction, a second setup (referred to as "setup 2" for the rest of the study) involving laser beam shaping was developed (Figure 3). Many advantages arise when shaping a laser beam intensity profile for laser ablation. First, it allows for the use of a working fluence that is very close to the material ablation threshold and thus maximizes resolution and minimizes overexposure. Second, it limits the fraction of the intensity profile that does not contribute to the ablation and that can generate a heat-affected zone (HAZ) around the ablation zone. Finally, it permits the control of the geometry of the ablation crater. These effects are illustrated in Figure 4 [17].

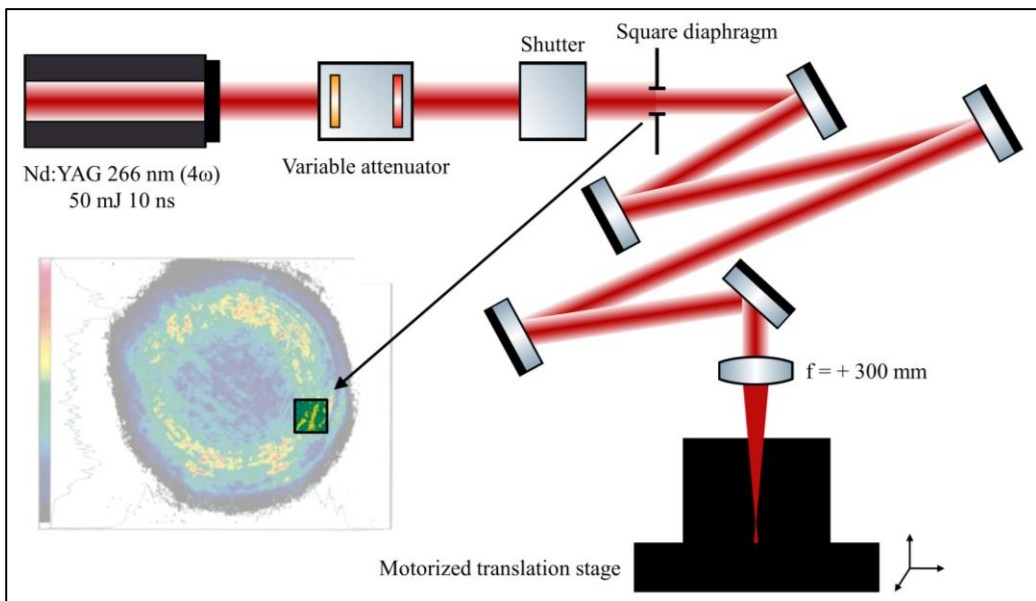

**Figure 3.** Second setup implemented for laser ablation of varnish. A square adjustable diaphragm was added to select a relatively homogeneous area of the intensity profile in the near field and create its image at the surface of the mock-up. Mirrors were added to increase the distance between the diaphragm and lens to minimize the size of the image. This feature allows for the performance of the laser ablation of the surface with a micrometric square top-hat beam.

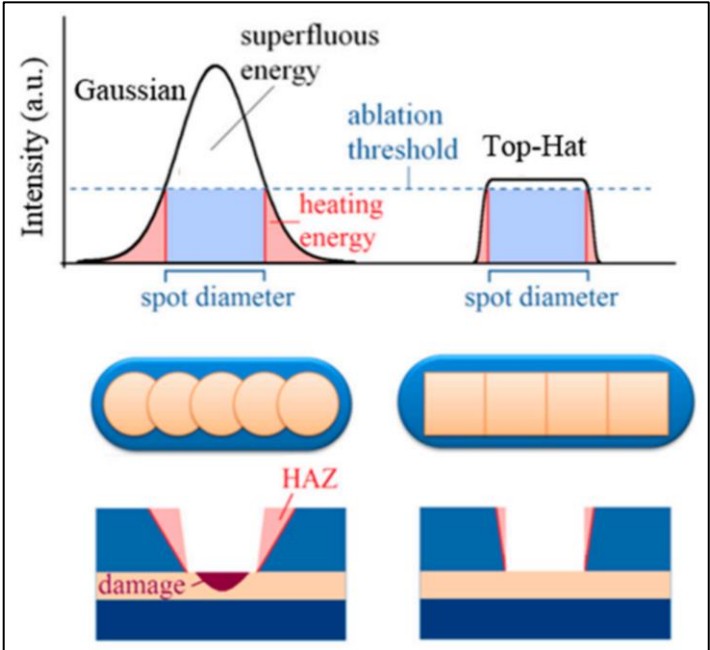

**Figure 4.** Illustration of the interests of beam shaping (taken from [17]). Most of the laser energy is used for ablation, HAZ and overexposure are limited, and geometry in interaction zone can be controlled.

The beam-shaping technique implemented in this setup is one of the most elementary among the many existing ones [18]. It consists of selecting a relatively homogeneous fraction of the light beam in the optical near field with an adjustable square mask added to the first setup and imaging the residual irradiance in the plane of the mock-up. To manage the magnification and thus obtain the desired beam size in the interaction zone, the distance between the mask and focusing lens is adjusted by folding the beam with multiple mirrors.

This method is fundamentally robust and simple to implement, but it is energy-inefficient because a significant fraction of the laser beam is intrinsically occulted in the process. However, given the low ablation threshold of dammar varnish [13] and the available laser power, the fluence was sufficient to induce the ablation under these conditions.

### 2.2. Monitoring and Control

2.2.1. Optical Coherence Tomography

OCT is an optical method that allows for the direct visualization of the stratigraphy of a multilayered object that is transparent or semitransparent at the OCT operating wavelength. It is based on the analysis of the light reflected by the internal structure of the object (at each interface where the optical index changes) [19]. The information obtained on the intensity of the back-reflection and its delay allows the reconstruction of a single-point signal (A-scan) that represents the location of the different interfaces along the optical path. The lateral displacement of the beam to adjacent positions allows the realization of a two-dimensional image of the internal structure of the object in the form of a section (B-scan). By acquiring B-scans side by side, it is possible to obtain a volumetric representation of the monitored area. This method is completely safe and noninvasive, and it has been widely studied for the analysis of cultural heritage artifacts [20–22]. It seems to be particularly useful for the measurement and control of varnish layers, and especially for the direct monitoring of laser varnish removal [23].

The OCT system used in this study consists of a Thorlabs GAN220-OCT base unit and Thorlabs OCTP900M scanner. The light source is a superluminescent diode ($\lambda_c$ = 912 nm; $\Delta\lambda$ = 803.8–1021.5 nm). It operates at 36 kHz with axial resolutions of 3 μm in air and 2 μm in varnish (for a typical varnish index of $n$ = 1.5).

2.2.2. Laser-Induced Luminescence

LIL is an optical spectroscopic technique that consists in exciting the molecular states of a material with a laser (below its alteration threshold) to analyze the luminescence re-emitted by the material as it returns to its ground state. The spectral characteristics of this re-emitted radiation are representative of the molecular nature of the material. LIL allows us to study the possible photochemical modifications induced in the residual varnish and pigments during the laser treatment by comparing the emission spectrum of the varnish–paint system (generally showing features related to the emissions of the varnish, pigments, and binder used for the painting) with the spectrum of the same system after ablation [4,12,13].

For these experiments, the laser excitation source was the same UV $4\omega$ ns Nd:YAG used for the ablation. It was coupled to a time-resolved fibered spectrometer Andor Shamrock 303i (Belfast, UK) (190–1100 nm, 600 grooves/mm) combined with an ICCD camera. Each measurement is the average of two acquisition points in two different locations of the sample, and each acquisition is the average of 1000 spectra. The acquisitions were triggered by a fraction of the pulses sent to a photodiode with a beam splitter.

### 2.3. Samples

For this study, representative samples of easel paintings were made. Canvases on frames were covered with preparatory layers resulting from a mixture of skin glue and calcium carbonate ($CaCO_3$). After drying, these primed canvases were painted and varnished.

The paint was prepared a tempera. Because of their high photosensitivity, the pigments chosen were red lead ($Pb_3O_4$, ref: 212571 Couleurs du Quai®, Paris, France), white lead (($PbCO_3)_2$-$Pb(OH)_2$, ref: 108 Sennelier®, Paris, France), and cobalt blue ($CoAl_2O_4$, ref: 0558 CTS®, Paris, France). The pigments were ground with water and mixed with a whole egg (white + yolk) in a 1:1 mass ratio. Water was then added until the desired consistency was obtained. In each case, three layers of paint were applied, with one week of natural drying between each layer.

Dammar varnish was chosen because it has been very popular in painting since the 19th century and has therefore been widely studied in the case of laser varnish removal [12,13,24]. For its preparation, 30 g of dammar resin in tears (Laverdure®, Paris, France) was placed in a nylon membrane filter, and the latter was placed in a jar into which 75 mL of turpentine was poured. After tightly recapping the jar, the resin was left to dissolve for 48 h at room temperature. The filter containing only pollutants and an insoluble fraction of resin was then removed, and the solution was transferred to another jar, leaving the waxy deposit at the bottom of the original jar. After drying the three layers of paint, three layers of varnish were applied at 24 h intervals. The samples were then placed in a custom artificial aging chamber.

The chamber is a wooden cabinet that is covered inside with a reflective aluminum coating. Light is provided by a combination of two fluorescent tubes:

- A Philips Master TL-D 90 De Luxe 58W fluorescent tube (Amsterdam, the Netherlands), allowing the simulation of the daylight at noon (color T° = 6500 °K);
- A Philips CLEO Performance 100 W fluorescent tube, emitting 99% of its radiation in the UVA range (between 310 and 400 nm) and 1% in the UVB range (between 280 and 315 nm) to simulate UV exposure in normal museum conditions.

The temperature and hygrometry in the chamber were measured with a thermometer and hygrometer, respectively. The temperature was stable around 47 °C, and the hygrometry (over which we had no control) was about 25%. Despite these conditions being relatively far from optimal museum conditions (a temperature of 21–22 °C and hygrometry between 45 and 55%), the samples were placed in this chamber for one month, and a visually satisfactory yellowed and cracked film of dammar resin was obtained.

Figure 5a shows the micrograph of a section of one of the painted mock-ups before the varnish application. Figure 5b shows an OCT B-scan of the same mock-up after varnishing, revealing the resolved thin film of dammar.

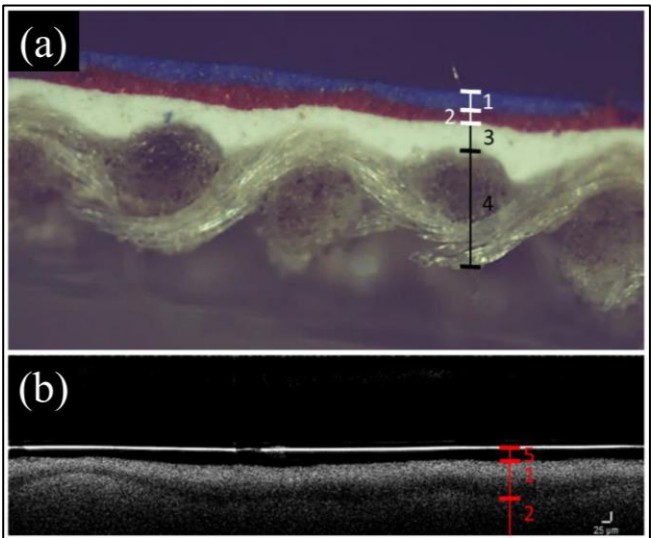

**Figure 5.** (**a**) Micrograph of section of painting mock-up showing (1) layer of cobalt blue, (2) layer of red lead, (3) preparatory layer, and (4) canvas. (**b**) OCT section showing (5) varnish layer, (1) layer of cobalt blue, and (2) layer of red lead.

Figure 6 shows the drift of the absorption range of dammar resin with ageing towards the near UV. Even though 4ω Nd:YAG radiation is less absorbed by dammar than 5ω, this absorption tends to increase with ageing. Just after its application, dammar resin is transparent at 4ω, but this transparency drastically decreases with ageing [17], making this wavelength theoretically relevant for the laser ablation of aged dammar varnish.

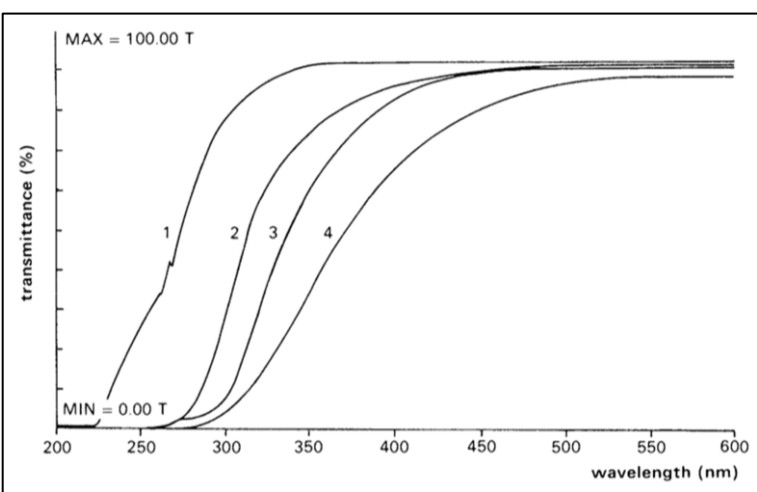

**Figure 6.** Transmittance spectra in UV–Vis (taken from [25]) of naturally aged dammar resin film behind south-facing window: (1) after application; (2) after 122 days; (3) after 329 days; (4) after 1288 days.

## 3. Results

### 3.1. Optimization of Laser Ablation

Preliminary experiments were performed on a small area of a mock-up composed of red lead paint coated with dammar varnish and artificially aged. Directly focusing the laser beam on the sample (setup 1, Figure 2) above the ablation threshold triggered material removal. In this case, the geometry of the ablation crater (Figure 7a) was roughly a footprint of the asymmetrical laser intensity profile in the far field (Figure 1b). In the next experiment, the translation stage was moved at different speeds while the laser was running to obtain lines of ablated material. Different overlapping rates of the laser spot during ablation and different numbers of passes were achieved.

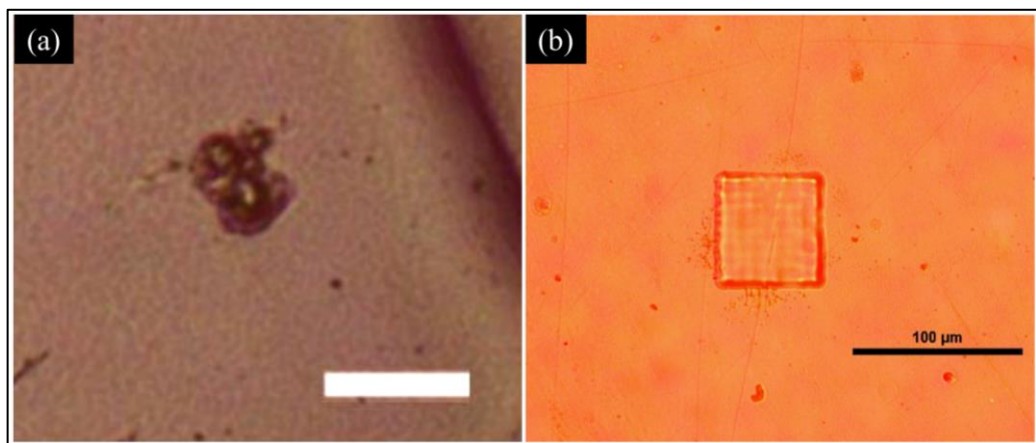

**Figure 7.** (**a**) Micrograph of single-shot ablation crater (~400 mJ/cm$^2$) obtained with setup 1 (without beam shaping) on varnish sample. White bar represents 200 μm. (**b**) Micrograph of single-shot ablation crater (~300 mJ/cm$^2$) obtained with setup 2 (with square beam shaping) on varnish sample.

Figure 8 shows a micrograph of the results. The footprint of the asymmetrical intensity profile resulted in a poorly defined and inhomogeneous ablated area surrounded by a slightly melted HAZ, which reduced the precision of the ablation process.

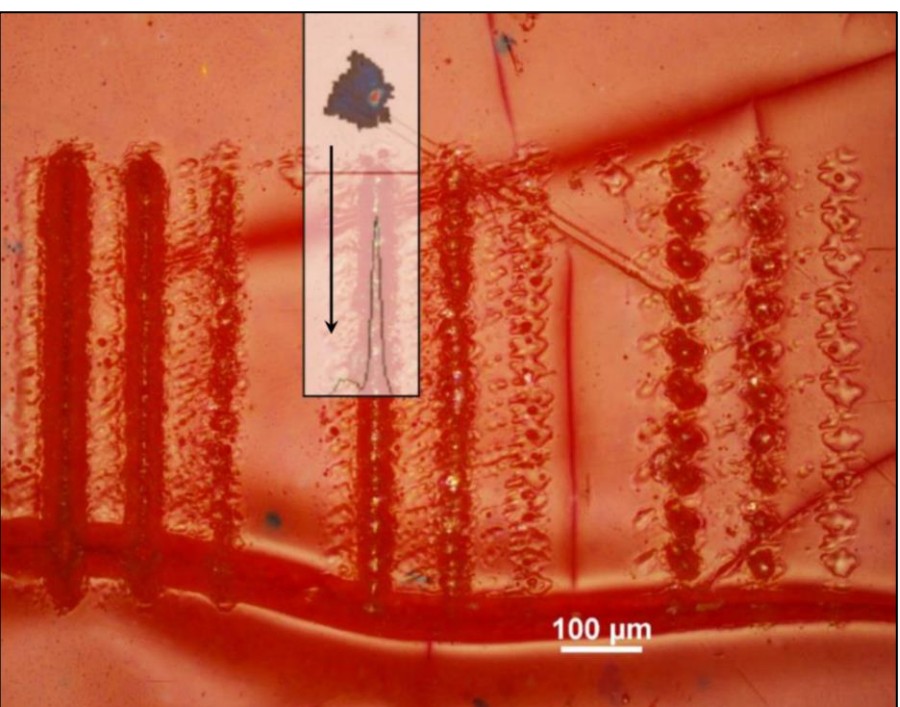

**Figure 8.** Lines of ablated dammar varnish (on red lead paint) obtained with setup 1. From left to right: 10, 5, and 1 pass with stage speed of 0.16 mm/s; 10, 5, and 1 pass with stage speed of 0.40 mm/s; 10, 5, and 1 pass with stage speed of 0.80 mm/s.

Setup 2 (Figure 3) was implemented, and similar preliminary tests were carried out. Figure 7b shows a micrograph of a single-shot crater obtained on the same sample. The geometry of the ablation crater was perfectly controlled and localized with setup 2, which was retained for the rest of the varnish removal study.

### 3.2. Overlapping Rate

Further tests consisted in extending the ablation process to a 2D surface by scanning the motorized stage at different speeds with setup 2.

We defined a parameter (*R*) for the overlapping rate that takes continuous values between 0% and 100%, depending on the speed of the motorized translation stage. For instance, the stage speed corresponding to a value of *R* equal to 100% means that the stage speed is zero, and the successive pulses are deposited on the same spot. A value of *R* equal to 0% means that the stage moves a distance exactly equal to the length of the ablation crater between two pulses, and the overlapping of the laser spot is thus zero for one laser pass. Any intermediate value of the stage speed corresponds to an intermediate overlapping rate (*R*).

When choosing an overlapping rate, it is important to take into consideration that the higher the overlapping rate, the more pulses are deposited on the same spot for one laser pass. Therefore, the amount of material removed per laser pass increases with the overlapping rate. Figure 9 shows a theoretical plot representing the number of pulses deposited on the same spot for one laser pass of 100 consecutive shots as a function of the overlapping rate (*R*). Another parameter to consider is the homogeneity of the ablated extended area as a function of the overlapping rate (*R*). The general rule is that if the result of the difference (*100-R*) is a divisor of 100, then the overlapping rate (*R*) will induce a homogeneously ablated area. Otherwise, the surface will not be ablated with the same number of pulses at each spot, which leads to an inhomogeneously ablated area.

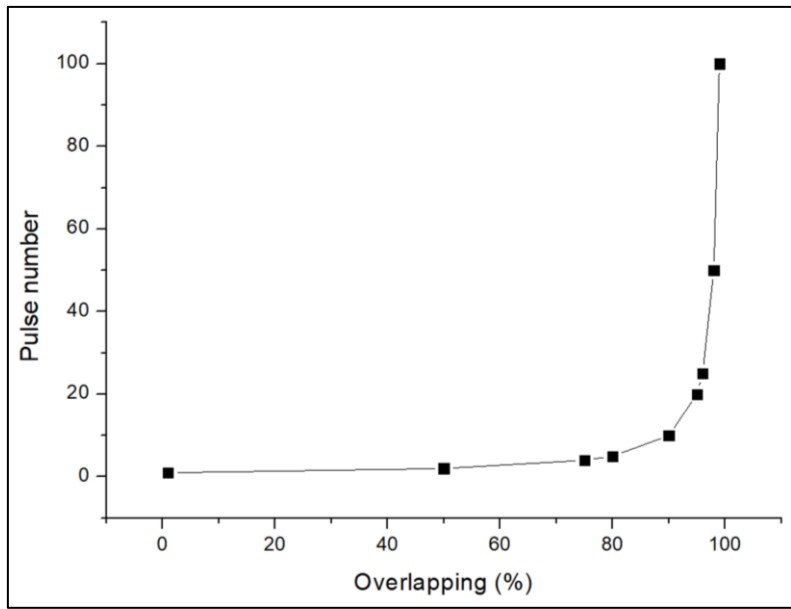

**Figure 9.** Plot showing number of pulses deposited on same spot for one laser pass of 100 consecutive shots, depending on overlapping rate (*R*). For *R* = 0%, a single shot is deposited on each spot. For *R* = 100%, the stage speed is null, and 100 shots are deposited on the same spot. Any intermediate stage speed implies an intermediate overlapping rate, and consequently, a different degree of irradiation.

Figure 10 presents a micrograph, an OCT volume acquisition, and an OCT B-scan of three ablation tests on an extended area. They were obtained at ~400 mJ/cm$^2$ on a sample of dammar-varnished red lead paint (Figure 10a,b) and a sample of dammar-varnished cobalt blue paint (Figure 10c). Different overlapping rates were used, and a schematic of the corresponding scan pattern shows, in color, the level of the homogeneity of the ablation and the number of pulses deposited (darker means more pulses). At 0% (Figure 10a) and 80% (Figure 10c) overlaps, the overlapping rate satisfies the condition for homogeneous ablation, and the number of pulses is equal in the area of interest (neglecting the edge effects of the ablated area). At 40% overlap (Figure 10b), the overlapping rate does not satisfy the homogeneous ablation condition, and the number of pulses is unequal in the ablated area, resulting in an unequal residual surface aspect.

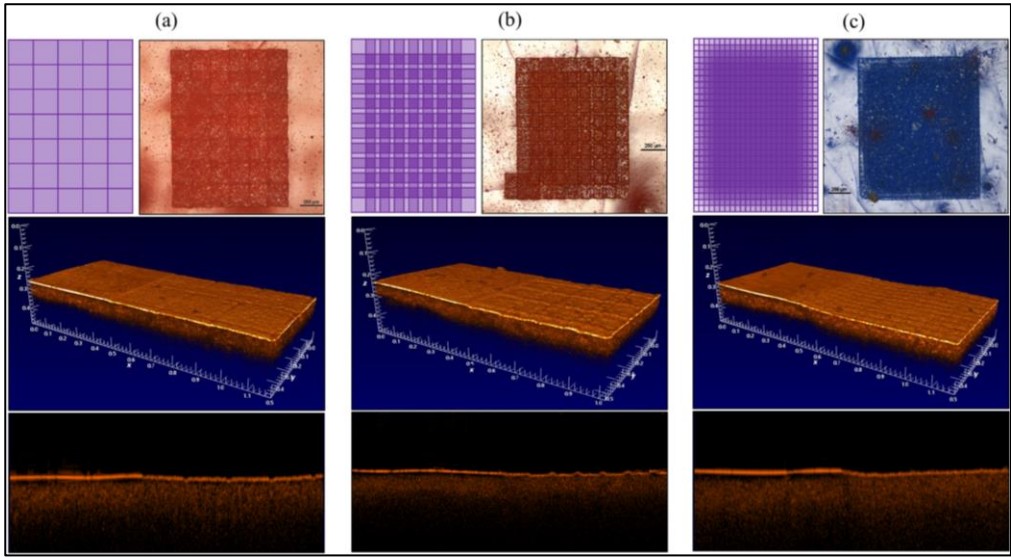

**Figure 10.** Scheme, micrograph, OCT volume acquisition, and OCT B-scan of three tests of ablation on extended area with (**a**) 0% overlap, (**b**) 40% overlap, and (**c**) 80% overlap.

### 3.3. Calibration of Laser Ablation of Varnish

To achieve controlled and gradual ablation, it is necessary to calculate the ablation rate (i.e., the amount of material removed per pulse as a function of the fluence), and then to determine the amount of material removed at a given fluence as a function of the number of pulses. Thresholds and ablation rates for dammar varnish are available in the literature [13,24], but their values can vary depending on the degree and condition of the ageing of the varnish. Therefore, this calibration procedure must be carried out on a case-by-case basis.

Calibration was performed with setup 2 on a mock-up of cobalt blue paint varnished with dammar and artificially aged. The first step was to create a crater matrix between 100 mJ/cm² and 1 J/cm². This is shown in the top image of Figure 11 (area B) (the black arrow going from high to low fluence). In order to decrease the statistical error, from five to six craters were made for each fluence value.

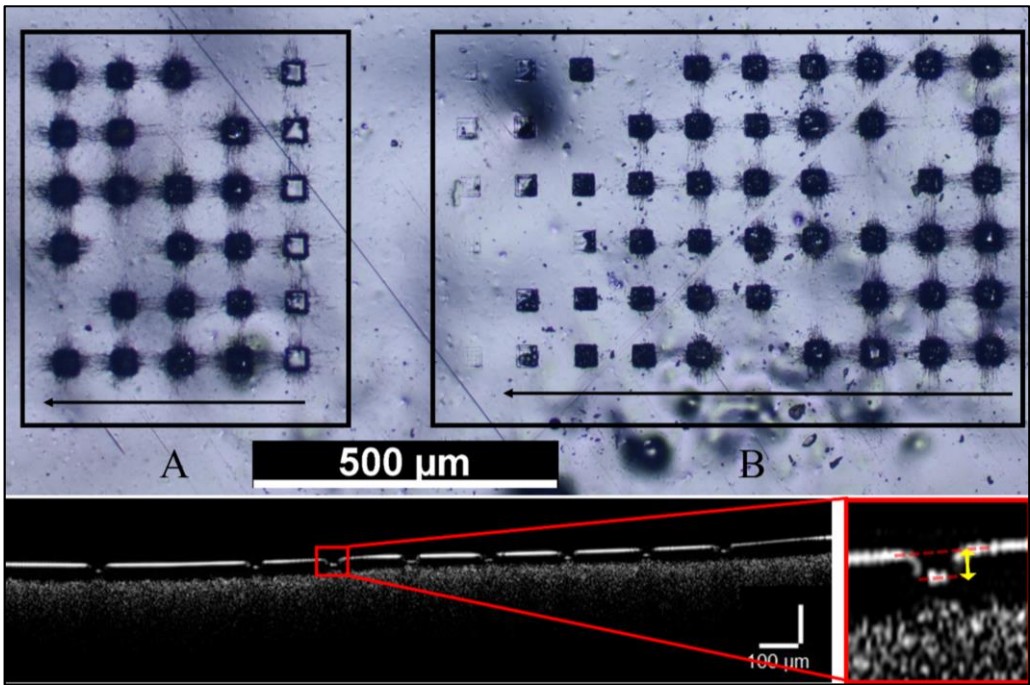

**Figure 11. Top**: Micrograph of calibration area on dammar-varnished cobalt blue mock-up with (**A**) various numbers of pulses at 1J/cm² (1, 5, 10, 15, and 20 pulses in direction of black arrow), and (**B**) craters obtained at various fluences (from 1 J/cm² to 100 mJ/cm² in direction of black arrow). **Bottom**: Example of OCT B-scan allowing measurement of crater depths. The yellow arrow represents the crater deapth measured.

To obtain measurable crater depths given the axial resolution of the OCT, each crater was the result of 5 pulses between 1 J/cm² and 600 mJ/cm², 10 pulses between 600 and 340 mJ/cm², and 15 pulses between 300 and 100 mJ/cm².

The crater depths were then measured by OCT and plotted as a function of the fluence. On this plot (Figure 12B), the error bars are 1 standard deviation on from 5 to 6 crater depths. The ablation rate data obtained were fitted to the "blow-off" model. The "blow-off" model is often used to predict the laser ablation rate of organic tissues and polymers in a moderate fluence range [14,25]. This very simple model requires several conditions to be valid. First, the Beer–Lambert law must correctly describe the spatial distribution of the laser energy absorbed in the material. Then, a threshold value of the energy density must exist to initiate the ablation, below which the irradiation only leads to the heating of the material. Another condition is that the ablation of the material is considered initiated at the end of the laser pulse. Finally, the conditions for ablation in the thermal confinement regime must be satisfied. These conditions are almost always met for the laser ablation of

organic materials in the ns regime [14]. Thus, for an incident fluence ($F_0$), the fraction of material exposed to a fluence above the threshold fluence ($F_{th}$) is ablated. This results in the semilogarithmic behavior of the ablation depth as a function of the fluence:

$$z_{ab} = \delta \ln \frac{F_0}{F_{th}}$$

where $z_{ab}$ is the ablation depth; $\delta$ is the optical penetration depth (equal to the inverse of the effective absorption coefficient ($\alpha_{eff}$) [14]); $F_0$ is the working fluence; $F_{th}$ is the ablation threshold. The saturation of the ablation rate theoretically occurs when $z_{ab} = \delta$; thus, when $F_0 = e \times F_{th}$. The fitting of the experimental data with the blow-off model allowed for the determination of the ablation threshold of the dammar varnish ($F_{th} \approx 170 \text{ mJ/cm}^2$) and optical penetration depth ($\delta \approx 3 \text{ μm}$) (i.e., $\alpha_{eff} = 0.33 \text{ μm}^{-1}$). The latter value is lower than that reported in the literature, which is in the range from 11 to 12 μm [13]. This can be explained by the differences in the experimental conditions that led to this value. The literature value was obtained by analyzing UV–Vis spectra (i.e., at a very low intensity), and it corresponds more to the inverse of the linear absorption coefficient. The value of the $\delta$ obtained from the experimental fit (i.e., at a fluence higher than the ablation threshold) indirectly integrates the complex mechanisms that are involved in ablation and therefore corresponds rather to the inverse of the effective absorption coefficient ($\alpha_{eff}$). Among these mechanisms, the example of the shielding of a fraction of the pulse by the dense cloud of ejected fragments (more absorbent than the varnish at this wavelength) has already been described in the literature [14] and can explain this saturation effect of the ablation rate at a depth lower than the theoretical optical penetration depth.

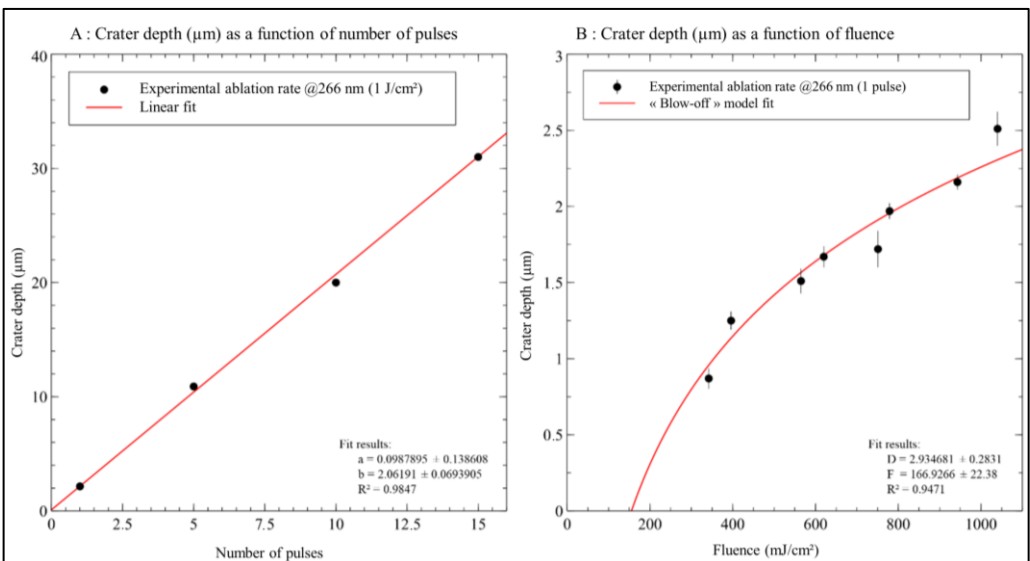

**Figure 12.** Plots showing results of calibration procedure with (**A**) crater depths as function of number of pulses for fixed fluence and (**B**) crater depths as function of fluence fitted with "blow-off" model.

In a second step, an array of craters was realized at 1 J/cm² with different numbers of pulses. The result is shown in Figure 11 (area A), with the black arrow pointing in the direction of the increasing number of pulses. In the same way as in the previous experiment, from five to six craters were made for each number of pulses, and the depths of the craters were measured and plotted as a function of the number of pulses. The standard deviation over from five to six measurements was computed, but the error bars are barely visible. This depth happened to be a linear function of the number of pulses (Figure 12A) and was possibly due to the low repetition rate and therefore the lack of a shielding effect and heat accumulation from one pulse to the next.

### 3.4. Controlled Laser Removal of Dammar Varnish

A controlled varnish removal was then carried out with setup 2 on a paint mock-up (dammar-varnished red lead mock-up). Considering the results of the calibration procedure, the working fluence was set at 800 mJ/cm² (~2 μm removed per pulse), with an overlapping rate in X of 80%, corresponding to five pulses per line and per laser pass, and a minimum overlapping rate in Y. Under these conditions, the removal of a 10 μm thick varnish layer was expected.

Figure 13 illustrates the results obtained. Figure 13a presents an OCT volume acquisition showing the transition between the original varnish and ablated area under these conditions. Due to the square beam shaping, the boundary between these two areas is very well defined, reflecting the highly localized nature of the laser treatment. The edges of each ablation line are visible due to the minimal overlap chosen in the direction perpendicular to the scanning direction. Figure 13b,c are the OCT B-scans from the volume acquisition (Figure 13a) confirming the homogeneity of the ablated area, as well as the depth achieved, and in agreement with the calibration performed, as approximately 10 μm of the varnish was removed.

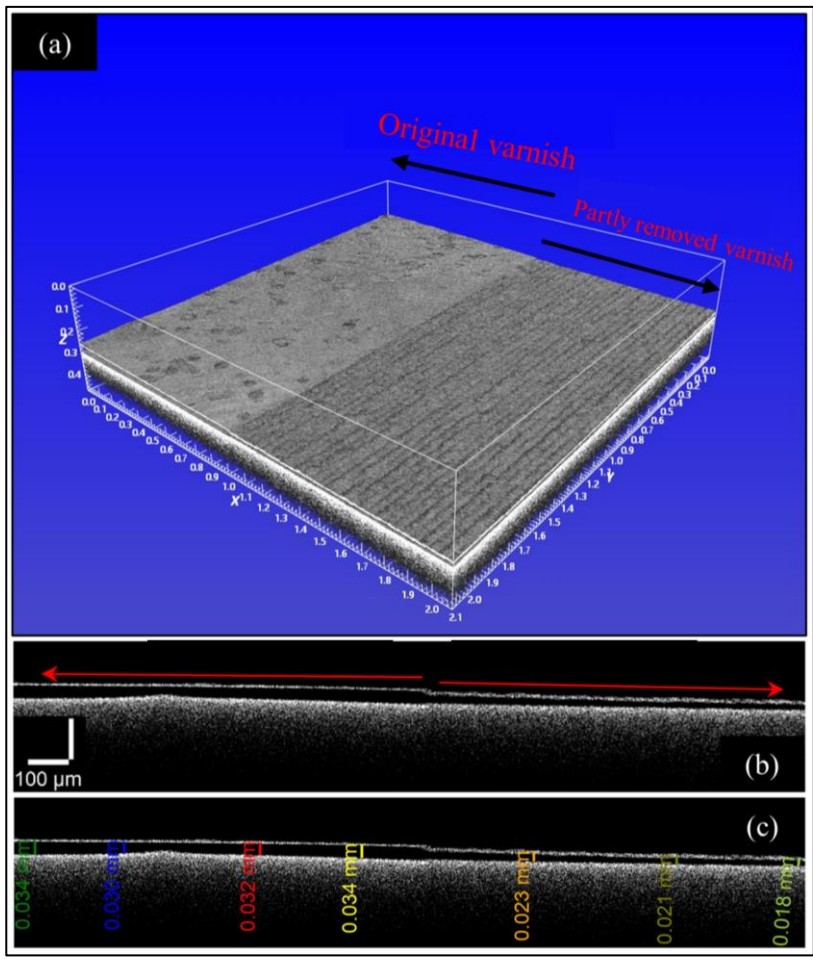

**Figure 13.** (**a**) OCT volume acquisition of transition between intact varnish and laser-ablated varnish with setup 2 at 800 mJ/cm². (**b**,**c**) Section (OCT B-scan) of same volume acquisition showing ablation frontier. Around 10 μm of varnish was removed.

### 3.5. Spectroscopic Assessment of Remaining Material

A laser-induced-luminescence study was then performed on a partially laser-ablated varnished red lead mock-up to check the integrity of the pigment colors and remaining varnish. Spectra were acquired on a nonablated varnish area to serve as a reference. Spectra

were then acquired in laser-ablated areas. The results are displayed in Figure 14, together with the OCT B-scans of the measured areas.

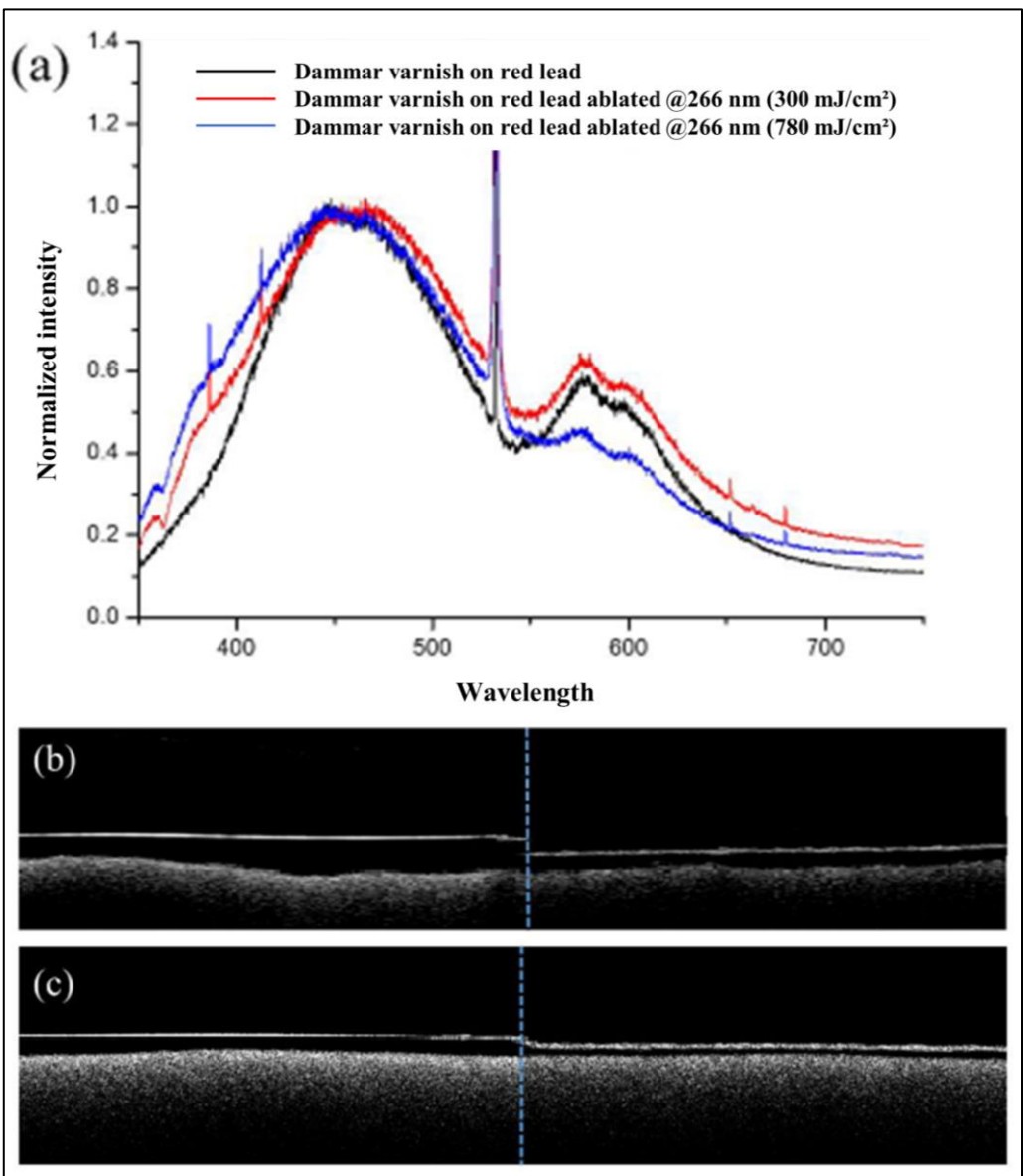

**Figure 14.** (**a**) Superimposed and normalized LIL spectra of dammar varnish on layer of red lead a tempera: black line: reference spectrum; red line: spectrum of ablated area at $4\omega$, 300 mJ/cm$^2$; blue line: spectrum of ablated area at $4\omega$, 780 mJ/cm$^2$. (**b**) B-scan of ablated area at $4\omega$, 300 mJ/cm$^2$. (**c**) B-scan of ablated area at $4\omega$, 780 mJ/cm$^2$.

To evaluate the influence of this parameter, laser ablation was performed on the same previous mock-up at two different fluences. In the first case, a fluence of 300 mJ/cm$^2$ was chosen, with $R = 80\%$ in the laser scan direction and $R = 80\%$ in the perpendicular direction. The samples were scanned once, resulting in 25 pulses per spot. The original varnish layer (although inhomogeneous) was about 50 μm thick, and about 20 μm was removed (Figure 14b), as expected from the calibration curve (Figure 12b). In the second case, a fluence of 780 mJ/cm$^2$ was used, with $R = 80\%$ in the scan direction and $R = 50\%$ in the perpendicular direction. The surface was scanned once, resulting in 10 pulses per spot. Under these conditions, approximately 25 μm of the varnish was removed (Figure 14c), as expected from the calibration procedure (Figure 12b).

The reference spectrum (Figure 14a, black line) is characterized by a broad emission band between 350 and 700 nm that is attributed to the luminescence of the dammar resin [12]. The broad band attributed to the dammar resin is accompanied by two narrower bands centered at about 580 and 610 nm. These bands are attributed to the red lead signature. The fluorescence spectrum of the residual varnish after ablation at $4\omega$ with a fluence of 780 mJ/cm$^2$ shows a slight redshift of the dammar signal, which can be explained by the difference in the oxidation state of the varnish after laser ablation [12]. Two shoulders at about 360 and 380 nm are also present, whereas they are absent in the reference spectrum. They were initially attributed to a possible photochemical modification of the material, but, in fact, it seems that these spectral features become more intense as the residual layer probed by LIL becomes thinner. Upon further investigation, these spectral contributions below 400 nm, which are more pronounced in the spectrum corresponding to a thinner residual varnish layer (Figure 14c), could be attributed to the luminescence of the egg white of the binder used in the paint, which is known to have a spectral response between 330 and 400 nm when excited by UV light [26]. Finally, the emission features of the pigments are aligned with those of the reference spectrum, indicating the preservation of their physicochemical integrity after ablation. However, it appears that their relative intensities are nevertheless lower. This is due to the opacification of the residual varnish following ablation at $4\omega$ and the scattering of the pigment emission signal, as discussed previously and in the next section. The fluorescence spectrum of the residual varnish after ablation with a fluence of 300 mJ/cm$^2$ shows the same spectral characteristics as those obtained on the areas ablated at a higher fluence, with two exceptions: the shoulders below 400 nm are less intense, and the bands attributed to the pigments show an intensity closer to that of the reference spectra. The first feature can be explained by a thicker residual varnish layer in this case (due to the relative inhomogeneity of the varnish layer), and thus a lower contribution of the egg-white fluorescence. The latter difference seems to be due to the better-preserved transparency of the residual varnish when ablated at a lower fluence, although this aspect was not studied in detail in this work.

In these conditions, the LIL showed that the photochemical integrity of the residual material seemed to be preserved during the laser ablation of the varnish at 266 nm ($4\omega$).

### 3.6. Opacification of Varnish after Laser Ablation

The physical appearance of the residual coating after ablation at $4\omega$ is the main argument against the use of this wavelength for varnish removal [13]. In fact, the generation of micrometric bubbles during ablation leads to a strong scattering of light and thus to the opacification of the varnish coating.

In the next experiment, three varnished areas were ablated (following the calibration procedure) in order to remove about 15 μm of dammar varnish. Micrographs of these areas are shown in Figure 15a,c,e. The upper part of each micrograph represents the original fraction of the varnish through which the pigment grains are visible, showing the transparency of the layer. The lower part of each image represents the ablated areas. A binocular image of an ablated area of varnish on red lead paint is also shown in Figure 16a. It allows for a more macroscopic rendering of the varnish blurring after ablation. The bubbled appearance of the varnish associated with a loss of transparency due to significant light scattering is clearly visible on the red lead (Figures 15c and 16a) and cobalt blue (Figure 15e) mock-ups.

Because the removal of varnish is often followed by the application of a new layer of varnish on the surface, a layer of synthetic varnish was applied. Laropal®A81 was chosen because it possesses similar optical properties to natural varnishes without posing the problem of yellowing over time [27].

The regeneration of the transparency was observed immediately after application. The varnish was then left to dry for four hours. Figure 15b,d,f presents micrographs of the ablated areas varnished with Laropal®A81 after laser treatment. It appears that the image of the pigment grains is sharp again through the layer, which suggests the regeneration of

the transparency. Figure 16b seems to confirm this effect, as the ablated area is not even visible after the application of the Laropal®A81 layer. These results look encouraging for the resolution of the bubble formation during varnish laser removal with $4\omega$, and they should motivate further discussions about the suitability of this wavelength for this application.

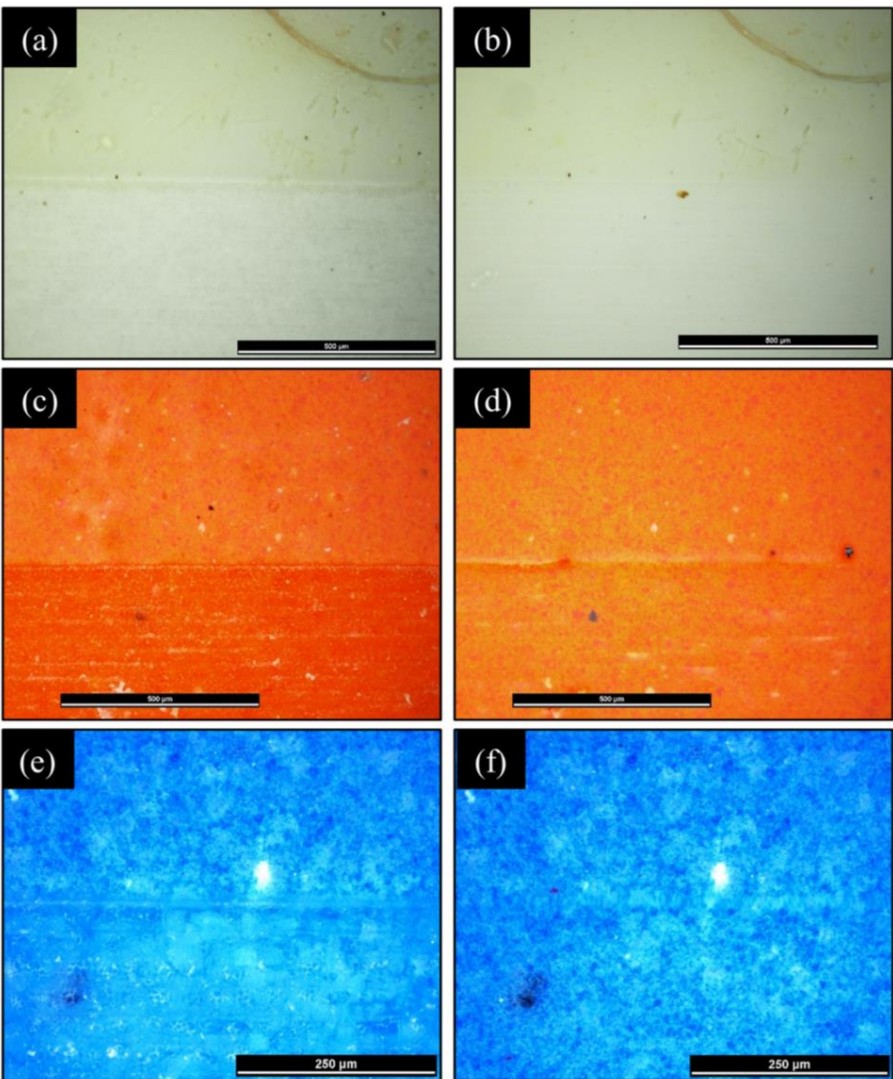

**Figure 15.** Micrographs of transition between original fraction of varnish and ablated one: (**a,b**) dammar varnish on white lead before and after varnishing ablated area with Laropal®A81, respectively; (**c,d**) dammar varnish on red lead before and after varnishing ablated area with Laropal®A81, respectively; (**e,f**) dammar varnish on cobalt blue before and after varnishing ablated area with Laropal® A81, respectively.

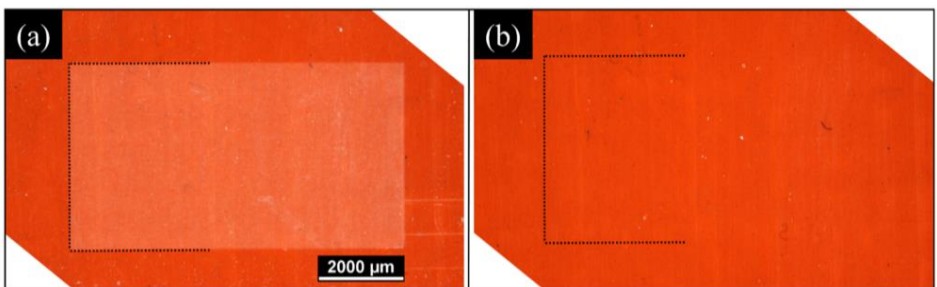

**Figure 16.** Binocular image of dammar-varnished red lead mock-up: (**a**) after ablation of varnish and (**b**) after application of Laropal®A81 layer on ablated area.

## 4. Conclusions

In this study, we proposed a comprehensive investigation of the laser removal of dammar varnish from easel paintings using a $4\omega$ ns Nd:YAG laser at 266 nm. The laser–matter interaction was optimized by shaping the intensity profile of the laser used. This resulted in a clear and well-defined square homogeneous beam for the precise ablation of the varnish surfaces. The overlapping rate during the ablation of an extended surface was also studied to obtain a homogeneous residual surface. A calibrated procedure for precise varnish ablation was then proposed. The procedure involves the use of OCT to measure the ablation rate. The data obtained, when fitted to the photoablative "blow-off" model, allow for the controlled micrometric removal of varnish. Laser-induced luminescence was performed on ablated varnished mock-ups to assess the photochemical integrity of the remaining material after laser ablation. Two sets of parameters (low and moderate fluences) were used to achieve the sufficient ablation of dammar varnish on photosensitive pigments. In both cases, the LIL revealed no modification of the remaining varnish, and no chromatic alteration of the paint. Finally, a solution was proposed to minimize the blurring effect on the remaining varnish after ablation at 266 nm. Following the regular conservation procedure, a synthetic varnish was applied to the areas of the mock-ups where the dammar had been ablated. The transparency was restored after the application of the Laropal®A81, and it seemed to be maintained after drying.

These results confirm the relevance of laser ablation for the sharp micrometric removal of dammar varnish from the surfaces of delicate paintings. This work also reevaluated the suitability of the 266 nm ($4\omega$) ns Nd:YAG laser for this application, and the results seem encouraging for the use of this source.

**Author Contributions:** Conceptualization, methodology, validation, investigation, and data curation, M.L. and V.D.; software and formal analysis, M.L. and X.B.; writing—original draft preparation and visualization, M.L., X.B. and V.D.; writing—review and editing, M.L., X.B., V.D. and N.W.-C.; supervision, V.D. and N.W.-C.; resources, project administration, and funding acquisition, V.D. All authors have read and agreed to the published version of the manuscript.

**Funding:** This research was funded by the French Ministry Research Program EquipEx PATRIMEX (ANR-EQPX-0034) and ESR ESPADON-PATRIMEX (ANR-21-ESRE-00050). The research work had the support of a grant under the Decree of the Government of the Russian Federation, No. 220 of 9 April 2010 (Agreement No. 075-15-2021-593 of 1 June 2021). Maxime Lopez thanks the Fondation des Sciences du Patrimoine (FSP) for funding his PhD thesis.

**Data Availability Statement:** Not applicable.

**Conflicts of Interest:** The authors declare no conflict of interest.

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
