# Peer review of "Contribution to Controlled Method of Varnish Removal from Easel Paintings by ns Pulsed Nd:YAG Laser"

_heritage, doi:10.3390/heritage6030175_

Round 1
Reviewer 1 Report
line 96: specific references needed
line 113: specific additional references needed
line 154: image reference needs a check: lack of correspondence between "3.1a" and "3.1b" (in the text) and "3.a" and "3.b", below the images.
line 187: image reference is different from that used in image 1 (name of the authors or just the number of the cited reference?)
lines 269-270: a more detailed description of the sample preparation should be usefull to better clarify the tests and the results obtained. What was the selected substare? Any preparation was applied below the painting layer? Was the varnish aged or not?
line 313: the cited references should be added
line 347: repetition of the word "predict"
line 348: repetition of the words "fluence range"
line 440: a more detailed description of the sample should be usefull
line 443: "aged" is referred to the varnish itself or to the ageing of the varnish after the application? In that case, what are the condition of the ageing treament?
line 464: The title seems to be in french
line 476: repetition of the words "images obtained"
line 478: check the formatting of the paragraph; repetition of the word "laser"
line 482: repetition of the words "reflectance spectra acquired"
line 485: repetition of the words "from yellow"
line 501: does any measurement and comparison was realised to quantify this result? Is was just supported by the experiences of the restorators?
line 502: scalpel are not generally used to remove varnish, and, in any case, the results of superficial opacification suggests that the treatment is to aggressive or not sufficiently effective. This is not generally accepeted as good result in the common practice. In this sense, this comparison does not support the idea that the opacification obtained after the laser test is acceptable.
line 530: Does the "four hours of drying time" refer to the layer of the Laropal or to the break between the laser cleaning and the application of the varnish? In case this refers to the drying of the varnish, the drying grade of the Laropal was somehow measured?
line 660: does the formation of these micrometric bubbles may result in further degradation phenomena of the superficial layers (e.i. micro-scalingbubbling of the varnish layer,..)? Does the following re-solubilisation of the bubbled varnish led to interferecences with the chemico-phyisical properties of the new varnish?
Reviewer 2 Report
In my opinion, the subject studied is interesting and makes a contribution to scientific knowledge, particularly the use of UV laser techniques applied to the conservation of artistic heritage.
The study design is strong in terms of its theoretical basis and is well supported experimentally. The paper is logically prepared and systematic in its presentation.
For my part, I recommend publishing it with some clarifications that are described below:
1- In materials and methods, line 93, it is said that the composition of the Dammar resin has been studied. On the other hand, in line 113 is mentioned that these resins turn yellow with age.
I miss some bibliographical reference in both cases.
2- In results, line 312, it is said that the thresholds and ablation rates for Dammar resin are available in the literature.
Also there I miss some bibliographical reference.
3- In results, line 388, it is said that an optical configuration has been developed especially for this work.
I understand that this configuration is based on the scheme of Figure 6, but I think it is necessary to explain and a little more the details of the configuration used in the scanning of the samples.
4- In results, line 356, it is said that the ablation threshold obtained for the Dammar resin is 155 mJ/cm². Later in line 444 it is said that, after the calibration procedure, the working fluence is set to 800 mJ/cm².
I think it is necessary to explain a little that calibration process to achieve that operation fluence.
5- In results, line 452, the minimum overlap chosen in the direction perpendicular to the scanning direction is mentioned.
I think it is necessary to quantify this overlap and explain the criteria for selecting it.
Reviewer 3 Report
The research work has novelty and shall find good application. The manuscript has few typo errors and misspelt words.
The figure 1 quality is poor, needs improvement.
Reviewer 4 Report
Please find comments annotated in the attached manuscript.
